# Trends in Pancreatic Cancer Incidence and Mortality in Lithuania, 1998–2015

**DOI:** 10.3390/ijerph19020949

**Published:** 2022-01-15

**Authors:** Povilas Kavaliauskas, Audrius Dulskas, Inga Kildusiene, Rokas Arlauskas, Rimantas Stukas, Giedre Smailyte

**Affiliations:** 1Laboratory of Cancer Epidemiology, National Cancer Institute, LT-08406 Vilnius, Lithuania; inga.kildusiene@nvi.lt (I.K.); giedre.smailyte@nvi.lt (G.S.); 2Laboratory of Clinical Oncology, National Cancer Institute, 1 Santariskiu Str., LT-08406 Vilnius, Lithuania; audrius.dulskas@nvi.lt; 3Clinic of Internal Medicine, Family Medicine and Oncology Institute of Clinical Medicine, Faculty of Medicine, Vilnius University, M. K. Čiurlionio Str. 21/27, LT-03101 Vilnius, Lithuania; 4Department of Public Health, Institute of Health Sciences, Faculty of Medicine, Vilnius University, LT-03101 Vilnius, Lithuania; rokas.arlauskas@mf.vu.lt (R.A.); rimantas.stukas@mf.vu.lt (R.S.)

**Keywords:** pancreatic cancer, incidence, mortality, age-standardized rates

## Abstract

Background: Pancreatic cancer is one of the deadliest cancers worldwide, and its incidence is increasing. The aim of this study was to examine the time trends in the incidence and mortality rates of pancreatic cancer for the period of 1998–2015 for the first time in Lithuania by sex, age, subsite, and stage. Methods: This study was based on all cases (deaths) of pancreatic cancer diagnosed between 1998 and 2015. Age-standardized incidence (mortality) rates and group-specific rates were calculated for each sex using the direct method (European Standard). TNM classification-based information reported to the cancer registry was grouped into three categories: (1) localized cancer: T1-3/N0/M0; (2) cancer with regional metastasis: any 1-3/N+/M0; (3) advanced cancer: any T/any N/M+. Joinpoint regression was used to provide annual percentage changes (APCs) and to detect points in time where statistically significant changes in the trends occurred. Results: Overall, 8514 pancreatic cancer cases (4364 in men and 3150 in women) were diagnosed and 7684 persons died from cancer of the pancreas. Pancreatic cancer incidence rates were considerably lower for women than for men, with a female:male ratio of 1:2. Incidence rates changed during the study period from 14.2 in 1998 to 15.0/100,000 in the year 2015 in men, and from 6.7 to 9.8/100,000 in women. Incidence rates over the study period were stable for men (APC = 0.1%) and increasing for women by 1.1% per year. Similarly, mortality rates increased in women by 0.9% per year, and were stable in men. During the study period, incidence and mortality rates of pancreatic cancer were close. For the entire study period, rates increased significantly in the 50–74 years age group; only cancer of the head of pancreas showed a decline by 0.9%, while tail and not-specified pancreatic cancer incidence increased by 11.4% and 4.51%, respectively. Conclusions: The increasing pancreatic cancer incidence trend in the Lithuanian population may be related to the prevalence of its main risk factors (smoking, obesity, physical inactivity, diet, and diabetes).

## 1. Introduction

In 2020, pancreas cancer was the 12th most common cancer with 495,000 new cases worldwide [1]. Almost half of new cases (47%) were registered in Asia and 28% in Europe. GLOBOCAN estimated that in 20 years, pancreatic cancer rates will increase by 70% with 844,000 new cases each year. Pancreatic cancer is one of the most devastating malignancies, with an incidence equaling mortality, and has the lowest 5-year survival proportion. The 5-year survival rate has increased throughout the years; for example, in the USA, it increased from 2.0% to 9.2% from 1975 to 2011. However, more than 50% of new cases are diagnosed with distant metastatic disease [2]. Risk factors for pancreatic cancer are smoking, obesity, physical inactivity, diet, inheritance, and genetic predisposition [2]. In Nordic European countries, the results do not differ much, and in the last 50 years, the 5-year survival has increased from less than 5% to a little less than 10%. This poor survival is due to several factors, including non-specific symptoms and anatomical location; because the pancreas is close to main vessels, less than 20% of tumors are operable when diagnosed [3]. If treatment is possible, surgery possesses a high mortality risk, which in high-volume centers can be around 1% [4]. Despite all the progress in diagnostic methods, surgical techniques and further treatment options, pancreatic cancer is a life-threatening disease affecting the whole world.

The aim of this study was to examine for the first time the timing and trends in the incidence and mortality rates of pancreatic cancer for the period of 1998–2015 in Lithuania by sex, age, subsite, and stage.

## 2. Materials and Methods

The number of incidents of cancer cases and the number of deaths were obtained from the National Cancer Registry. Mortality data from malignant neoplasms of the pancreas were extracted from the World Health Organization (WHO) Mortality Database by 5-year age group. The study was based on all cases (deaths) of pancreatic cancer diagnosed between 1998 and 2015 (International Classification of Disease—Oncology, 3rd edition, site codes C250–C259, all histology forms with behavior code 3). The Lithuanian Cancer Registry is a population-based cancer registry containing personal and demographic information (place of residence, sex, date of birth, and vital status), as well as information on diagnosis (cancer site, date of diagnosis, and method of cancer verification) and death (date of death, cause of death) of all cancer patients in Lithuania, where the population size was around 3 million residents according to the 2011 census. Lithuanian data on cancer incidence for decades have been included in the Cancer Incidence in Five Continents, a longstanding collaboration between the International Agency for Research on Cancer and the International Association of Cancer Registries, which serves as a unique source of cancer incidence data from high-quality, population-based cancer registries around the world [5]. Age-standardized incidence and mortality rates and group-specific rates were calculated for each sex, using the direct method (European Standard). Corresponding population data, by age, sex, and year were available from Statistics Lithuania. A more detailed analysis by age and stage of disease was performed. During the study period, the TNM system was used for coding the stage of disease. TNM-classification-based information reported to the cancer registry was grouped into three categories: (1) localized cancer: T1-3/N0/M0; (2) cancer with regional metastasis: any 1-3/N+/M0; (3) advanced cancer: any T/any N/M+. Subsites were grouped in the following categories: head of pancreas (C250), body of pancreas (C251), tail of pancreas (C252), and other sites (C259, pancreas, not otherwise specified; C258, overlapping lesion of pancreas; C257, other specified parts of pancreas; C253, pancreatic duct; and C254, islets of Langerhans).

Joinpoint regression was used to provide annual percentage changes (APCs) and to detect points in time where statistically significant changes in the trends occurred. Those statistically significant points in time are shown in Table 1. The joinpoint regression analysis identified the best-fitting points (joinpoints) where a significant change in the linear slope (on a log scale) of the trend was detected. For the tests of significance, we used a Monte Carlo permutation method. Annual per cent changes were considered statistically significant if *p* < 0.05. A maximum number of three joinpoints was allowed for estimations. Joinpoint analysis was performed for all ages combined and age-specific rates for the following age groups: less than 50, 50–74, and 75 years and more, for each subsite and stage of pancreatic cancer. Joinpoint software version 4.3.1.0 was used (National Cancer Institute, Bethesda, MD, USA).

Ethical approval for the analysis of the population-based cancer registry data was not required.

## 3. Results

During the study period in Lithuania, 8514 pancreatic cancer cases (4364 in men and 4150 in women) were diagnosed and 7684 persons died from cancer of the pancreas. Pancreatic cancer incidence rates were considerably lower for women than for men, with a female:male ratio of 1:2. Incidence rates changed during the study period from 14.2/100,000 in 1998 to 15.0/100,000 in the year 2015 in men, and from 6.7 to 9.8/100,000 in women (Figure 1). From 1998 to 2015, the incidence and mortality rates of pancreatic cancer were close (Figure 2).

The results of the joinpoint analysis of incidence time trends are shown in Table 1. Incidence rates over the study period were stable for men (APC = 0.1%) and increasing for women by 1.1% per year (Figure 1). Similarly, mortality rates increased in women by 0.9%, and were stable in men (Table 1, Figure 2). Age-group analysis showed that for the entire study period, rates increased significantly in the 50–74 years age group (by 0.9%), while the observed changes in the youngest and oldest groups were insignificant (Figure 3).

No significant changes were found in joinpoint analysis by stage of disease during the whole study period; a statistically significant change in trend was found for distant disease in the period 1998–2011, when incidence decreased by 5.5% per year, and for disease with a not-specified stage (APC of 2.3% for the period 2000–2015) (Table 1, Figure 4). Incidence changes during the whole study period by subsite are shown in Figure 5. Only cancer of the head of pancreas showed a decline by 0.9%, while tail and not-specified pancreatic cancer incidence increased by 11.4% and 4.51%, respectively.

## 4. Discussion

In Lithuania, the pancreatic cancer incidence rate increased by 0.6% annually and reached 9.6/100,000 in 2015. This was not an uncommon phenomenon, as cancer incidence rates increased worldwide. In 2020, the most affected countries were Hungary, Uruguay, Slovakia, the Czech Republic, Japan, and Austria, with incidences rate ranging from 9.0 to 11.2/100,000. Lithuania’s values quite closely match those these countries, and in a 5-year period, the most affected country list did not change significantly [6]. Furthermore, it was proven that during the last 10 years, age-adjusted incidence rates have increased in North America, Western Europe, and Oceania [7].

Important risk factors affecting pancreatic cancer development are smoking, obesity, physical inactivity, diet, non-O blood group, and diabetes [2,8]. In the last 15 years, the use of tobacco in Lithuania has diminished by 5.6%. Smoking rates among men also decreased from 42% to 29.9%. However, the percentage of women smokers was almost the same, 9.7% [9]. The same trends can be seen in the USA, where smoking decreased by 6.5% in nine years [10]. Heavy smoking can increase the risk of chronic pancreatitis more than three times [11]. Furthermore, smokers have a 2.2 times higher risk of pancreatic cancer development than non-smokers [12].

It has been proven that the consumption of fatty foods on a regular basis increases the risk of pancreatic cancer. Studies have revealed that obese or overweight men and women are at higher risk of dying from pancreatic cancer. There is evidence that excessive meat intake may also increase the risk of pancreatic cancer [13,14,15]. The assessment of the body mass index (BMI) of the Lithuanian population has found a normal body weight in only less than half (43.6%) of adults and in less than one in three (26.7%) of the elderly population, while obesity has been observed in more than one-third (35.7%) of adults and 40.7% of the elderly: almost one in five adults and almost one in three elderly people are obese, and 1.7% are underweight [16]. A comparison of BMI between men and women showed more women than men are either obese or underweight. With the increasing age of both men and women, the normal body weight decreases, whereas the number of individuals that are overweight and obese increases [13]. The number of people with a body mass index (BMI) greater than 30 increased from 16.0% in 2005 to 18.9% of the total Lithuanian population in 2019 [13]. If the current trends in obesity continue, in 2030 38% of the world population will be overweight and 20% of them will be obese [14]. It is still unclear exactly how obesity increases the risk of pancreatic cancer development; however, it is thought that it may be related to inflammation and hormonal imbalances [17]. A meta-analysis from 2003 showed that obese people (with BMI more than 30) are associated with an 1.19 to 1.43 times increased risk of pancreatic cancer [18,19]. Yet, obesity is a modifiable risk factor with effects in many areas of health, so it should be addressed accordingly.

Different diets are popular throughout the world. Anderson et al. [20] assessed the links between carcinogens formed in cooked meat and the incidence of pancreatic cancer. During the median follow-up of 10 years, the authors found that mutagens formed during the cooking of meat significantly increased the risk of pancreatic cancer. In 2014, the great majority of the Lithuanian population (78.6%) used oil for cooking, 11.8% used butter, 6.1% used margarine, and 2.9% used animal fats [21]. So far, there have been no significant differences observed in the types of fat used by the population for cooking since 1997. In 2019, about one-third (35.7%) of the Lithuanian population consumed meat and meat products every day, and the majority (44.2%) consumed meat and meat products 3–5 times a week [21]. Only 2.4% of the population did not eat meat and meat products. Type 2 diabetes mellitus is a known risk factor of pancreatic cancer [22]. The pathophysiologic derangements that are responsible for the development of diabetes mellitus have also been associated with an increased risk of cancer development [23]. Previously, we showed that a higher survival rate from diabetic pancreatic cancer was observed in groups of patients using metformin, dual therapy of metformin and antihyperglycemic medication combinations, or metformin and sulfonylurea combinations [24].

Hereditary factors also may affect development of pancreatic cancer. Pancreatic cancer patients within the non-A blood type group may have better long-term survival rates [25]. Furthermore, statistically significant differences were found when assessing blood types among pancreatic cancers, showing that the most common blood type was A [26]; however, authors agreed that further investigations are needed. Lastly, 5–10% of pancreatic cancers are inheritable, and some genetic disorders are associated with it [6].

In 2020, the International Cancer of the Pancreas Screening Consortium recommended pancreatic surveillance for high-risk patients selected by multidisciplinary teams in high-expertise centers [27]. This surveillance includes expensive and complex investigations including a magnetic resonance imaging (MRI) scan, endoscopic ultrasound, and blood glucose and serum CA 19-9.

Despite all the worldwide efforts, pancreatic cancer is a developing burden affecting all nations. In the absence of effective and efficient diagnostic and therapeutic tools, it is important to focus on prevention by eliminating modifiable risk factors associated with pancreatic cancer. Moreover, further research on etiological factors and related mechanisms is urgently needed.

The major strength of our study is the fact that we included all the population Cancer Registry data of Lithuania. For the first time, we were able to demonstrate trends in both the mortality and incidence of pancreatic cancer for the whole population of Lithuania. In addition, incidence trends by sex and age were analyzed. Our study obviously has some limitations. Firstly, a true increase may underlie the trend in incidence and mortality being reported. The fact that both incidence and mortality increased supports a non-artefactual effect. Secondly, comorbidities, diet, drug use, and other cofactors were not included in the analysis.

In conclusion, pancreatic cancer incidence and mortality rates are increasing for both sexes. For the entire study period, rates increased significantly in the 50–74 years age group; only cancer of the head of pancreas showed a decline, while tail and not-specified pancreatic cancer incidence increased. The increasing incidence trend may be related to the prevalence of major risk factors (smoking, obesity, physical inactivity, diet, and diabetes).

## Figures and Tables

**Figure 1 ijerph-19-00949-f001:**
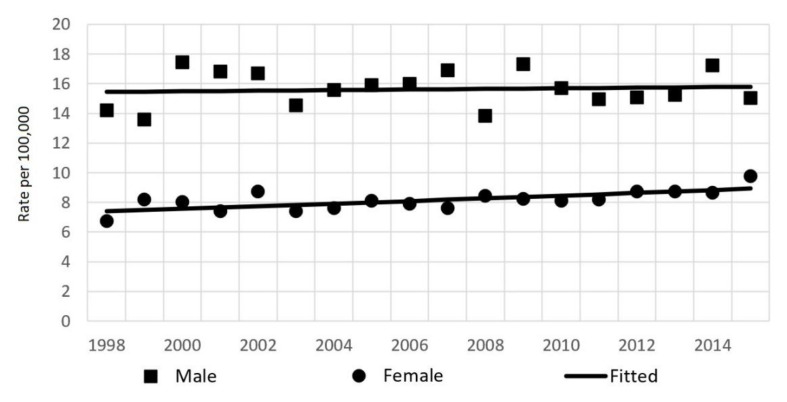
Age-standardized incidence rates of pancreatic cancer by sex in Lithuania from 1998–2015.

**Figure 2 ijerph-19-00949-f002:**
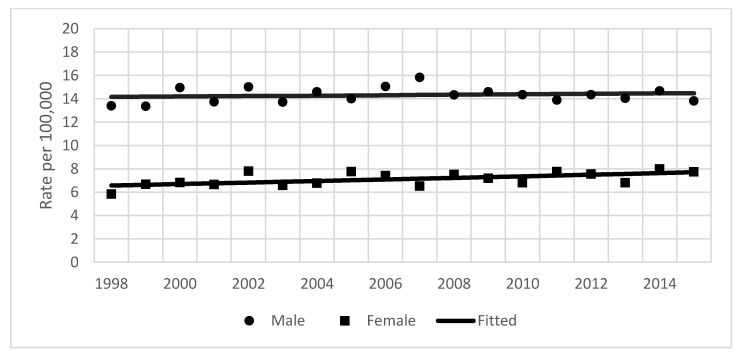
Age-standardized mortality rates from pancreatic cancer by sex in Lithuania from 1993–2012.

**Figure 3 ijerph-19-00949-f003:**
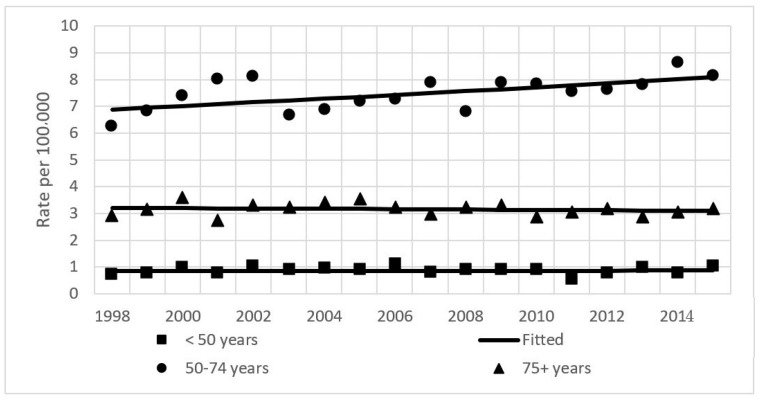
Age-standardized incidence rates of pancreatic cancer by age group in Lithuania from 1998–2015. Both sexes.

**Figure 4 ijerph-19-00949-f004:**
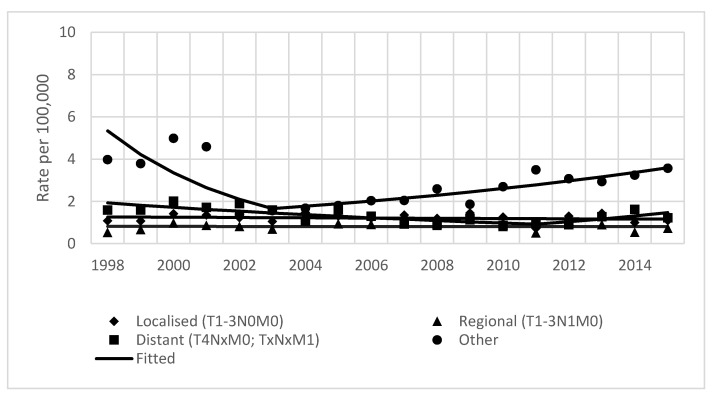
Age-standardized incidence rates of pancreatic cancer by stage of disease in Lithuania from 1998–2015. Both sexes.

**Figure 5 ijerph-19-00949-f005:**
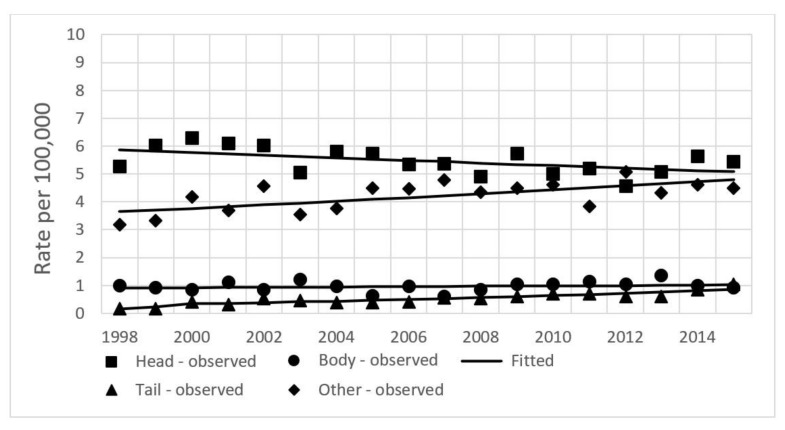
Age-standardized incidence rates of pancreatic cancer by subsite in Lithuania from 1998–2015. Both sexes.

**Table 1 ijerph-19-00949-t001:** Pancreatic cancer incidence and mortality joinpoint analysis in Lithuania from 1998–2015.

	1998	2015	APC	*p*	Trend 1	APC	*p*	Trend 2	APC	*p*
Variable	Rate	Rate	1998–2015							
Incidence										
Total	9.61	11.91	0.6	<0.0001	-	-	-	-	-	-
Sex										
Male	14.18	15.0	0.1	0.7	-	-	-	-	-	-
Female	6.72	9.79	1.1	<0.0001	-	-	-	-	-	-
Age										
<50 years	0.73	1.05	0.1	0.9	-	-	-	-	-	-
50–74 years	6.26	8.15	0.9	<0.001	-	-	-	-	-	-
75+ years	2.91	3.19	−0.2	0.5	-	-	-	-	-	-
Subsite										
Head	5.25	5.45	−0.9	<0.001	-	-	-	-	-	-
Body	1.00	0.91	0.7	0.4	-	-	-	-	-	-
Tail	0.15	1.05	11.4	<0.001	1998–2000	58.3	0.1	2000–2015	6.3	<0.001
Other	3.21	4.51	1.6	<0.001	-	-	-	-	-	-
Stage										
Localized (T1-3N0M0)	1.09	1.1	−0.5	0.5	-	-	-	-	-	-
Regional (T1-3N1M0)	0.54	0.74	0	1	-	-	-	-	-	-
Distant (T4NxM0; TxNxM1)	1.6	1.23	−1.6	0.5	1998–2011	−5.5	<0.001	2011–2015	12.4	0.2
Other	3.98	3.57	−2.3	<0.1	1998–2003	−20.8	0.01	2003–2015	6.6	0.01
Mortality										
Total	8.74	10.17	0.5	0.023	-	-	-	-	-	-
Sex										
Male	13.4	13.83	0.1	0.5	-	-	-	-	-	-
Female	5.86	7.77	0.9	0.009	-	-	-	-	-	-

## Data Availability

Data are available upon reasonable request.

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
