# Peer review of "Trends in Pancreatic Cancer Incidence and Mortality in Lithuania, 1998–2015"

_ijerph, 2022, doi:10.3390/ijerph19020949_

Round 1
Reviewer 1 Report
Kavaliauskas et al. investigated trends in the incidence and mortality of pancreatic cancer in Lithuania from 1988-2015. They observed increasing trends in both incidence and mortality rates in the entire Lithuanian population, as well as within specific subgroups.
Below are specific comments/limitations:
Background – It wasn’t mentioned until the discussion that this is the first study to evaluate pancreatic cancer trends in Lithuania. It would be helpful to include this information in the background. In general, there should be more emphasis in the background regarding the novelty of this study.
Background - The authors do a great job describing trends in risk factors in the discussion, but a brief description of pancreatic cancer risk factors should also be included in the introduction.
Methods – The term “Age standardized incidence (mortality) rates” is confusing to me. From the results it seems that the authors calculated both incidence and mortality rates. In that case, they can just specify “age standardized incidence and mortality rates” instead of putting “mortality” in parentheses.
Methods – Provide a reference or a brief description of Statistics Lithuania for non-Lithuanian readers who may not be familiar with this resource.
Figures – Missing the “- observed” suffix in the legend for Figure 2. Also check to make sure that the labeling of this suffix is consistent throughout the figures. In figures 1 and 3, it is labeled using both a capitalized and non-capitalized O.
Figures – Figure 4 is really hard to see because of the overlapping lines. Is there a way to change the shade of the colors or make the lines thinner so they don’t cover the shape of the data points?
Figures/tables – The data shown in the figures and tables are inconsistent. Figures 2-5 show trends in incidence rates for age, stage, and subsite, but there is no corresponding information in the table. The table instead shows trends in the mortality for these groups.
Results/tables – Include confidence intervals for the incidence/mortality rates and APCs in both the tables and the text
Table – Table is mislabeled as table 2. It would also be helpful to include total counts and percentages of each of these demographics and tumor characteristics. For instance, I cannot tell from this table how many cancers were diagnosed <50, 50-74, etc.
Table – Why are there different trend periods only for specific subgroups (e.g. tail, distant stage, other stage)? This needs to be described in the methods.
Discussion – The authors provide chi-square statistics for trends in several risk factors. Are these all taken from Statistics Lithuania? If so, this needs to be mentioned or cited.
Discussion – The discussion is mostly a description of trends in pancreatic cancer risk factors. While this is very comprehensive, it does not really interpret the results in relation to changes in risk factors. For instance, the authors should comment on whether the significant APC for females is due to increases in specific risk factors for females.
Discussion – Do the authors have a potential theory on why rates are increasing in the 50-74 group but not the other age groups?
Typos/grammatical mistakes – This manuscript could benefit from proofreading from a native English speaker to clean up grammar and syntax. I noticed several typos and grammatical errors (e.g. missing the word “the” before “study period” in a few sentences).
Author Response
Reviewer Nr. 1
Comment: Background – It wasn’t mentioned until the discussion that this is the first study to evaluate pancreatic cancer trends in Lithuania. It would be helpful to include this information in the background. In general, there should be more emphasis in the background regarding the novelty of this study.
Response: Changed as per suggestion.
Comment: Background - The authors do a great job describing trends in risk factors in the discussion, but a brief description of pancreatic cancer risk factors should also be included in the introduction.
Response: Changed as per suggestion.
Comment: From the results it seems that the authors calculated both incidence and mortality rates. In that case, they can just specify “age standardized incidence and mortality rates” instead of putting “mortality” in parentheses.
Response: Changed as per suggestion.
Comment: Methods – Provide a reference or a brief description of Statistics Lithuania for non-Lithuanian readers who may not be familiar with this resource.
Response: For this study we used data from the Lithuanian Cancer Registry, not Statistics Lithuania. Description of the Registry is extended in the Methods section
Comment: Figures – Missing the “- observed” suffix in the legend for Figure 2. Also check to make sure that the labeling of this suffix is consistent throughout the figures. In figures 1 and 3, it is labeled using both a capitalized and non-capitalized O.
Response: Charts edited as per suggestion.
Comment: Figures – Figure 4 is really hard to see because of the overlapping lines. Is there a way to change the shade of the colors or make the lines thinner so they don’t cover the shape of the data points?
Response: Figure 4 edited as per suggestion
Comment: Figures/tables – The data shown in the figures and tables are inconsistent. Figures 2-5 show trends in incidence rates for age, stage, and subsite, but there is no corresponding information in the table. The table instead shows trends in the mortality for these groups.
Response: We have corrected Table 1 to avoid misunderstanding of the data presented
Comment: Results/tables – Include confidence intervals for the incidence/mortality rates and APCs in both the tables and the text
Response: Confidence intervals are not provided, because there were data for whole population presented. They are informative for description of sample data only.
Comment: Table – Table is mislabeled as table 2. It would also be helpful to include total counts and percentages of each of these demographics and tumor characteristics. For instance, I cannot tell from this table how many cancers were diagnosed <50, 50-74, etc.
Response: Table number changed as per suggestion. Total counts are excess information, because for comparison between different countries only standardized rates are viable option.
Comment: Table – Why are there different trend periods only for specific subgroups (e.g. tail, distant stage, other stage)? This needs to be described in the methods.
Response: : In this sudy Joinpoint regression was used to provide annual percentage changes (APC) and to detect points in time where statistically significant changes in the trends occurred. This is already described in the Methods. Different trend periods were detected only for specific subgroups and they are presented
Comment: Discussion – The authors provide chi-square statistics for trends in several risk factors. Are these all taken from Statistics Lithuania? If so, this needs to be mentioned or cited.
Response: Discussion section has been changed according Reviewer 2 comment and now chi-square statistics are not provided
Comment: Discussion – The discussion is mostly a description of trends in pancreatic cancer risk factors. While this is very comprehensive, it does not really interpret the results in relation to changes in risk factors. For instance, the authors should comment on whether the significant APC for females is due to increases in specific risk factors for females.
Response: This study is a demonstration of trends Lithuania population. Our findings are complementary and do not differ a lot from other country data. Our studies design do not intend to analise risk factors for specific patient group. We noted that our studies limitation is that cofactors are not included in the analysis.
Comment: Discussion – Do the authors have a potential theory on why rates are increasing in the 50-74 group but not the other age groups?
Response: In other age groups incidence changes were insignificant and the rates are rather small. In this case is difficult to fix and interpret changes in trends
Comment: Typos/grammatical mistakes – This manuscript could benefit from proofreading from a native English speaker to clean up grammar and syntax. I noticed several typos and grammatical errors (e.g. missing the word “the” before “study period” in a few sentences).
Response: The publication was edited by a native speaker.
For editted manuscript, please see the attachment.
Reviewer 2 Report
This study investigated the trends in epidemiology of pancreatic cancer in Lithuania, and this might become a landmark article for the future investigation of pancreatic cancer in this country. There are several comments.
- Provide the reference of National Cancer Registry in this study (articles, annual report, or website, etc.). This is quite important step to confirm the reliability of the source of data. Pancreatic cancer included only ductal adenocarcinoma? How about other entities such as IPMC, PanNET, etc? State clearly about this issue.
- Table 2 might be Table 1 (Page 4).
- In discussion, there are lots of description regarding change in food as risk for pancreatic cancer. Decrease the sentence volume of this topic, and increase volume of other topics with more detailed considerations in the Lithuania`s epidemiology. Third paragraph of page 6 (line 170-172) in discussion section is not required.
- Fulfill the information regarding author contribution, findings, COI, acknowledgments, etc. (page 7).
- English should be edited by native speaker who are familiar with this field.
Author Response
Comment: Provide the reference of National Cancer Registry in this study (articles, annual report, or website, etc.). This is quite important step to confirm the reliability of the source of data. Pancreatic cancer included only ductal adenocarcinoma? How about other entities such as IPMC, PanNET, etc? State clearly about this issue.
Response: Reference provided and explanation on histology of tumours is included in Methods section
Comment: Table 2 might be Table 1 (Page 4).
Response: Changed as per suggestion.
Comment: In discussion, there are lots of description regarding change in food as risk for pancreatic cancer. Decrease the sentence volume of this topic, and increase volume of other topics with more detailed considerations in the Lithuania`s epidemiology. Third paragraph of page 6 (line 170-172) in discussion section is not required.
Response: Changed as per suggestion.
Comment: Fulfill the information regarding author contribution, findings, COI, acknowledgments, etc. (page 7).
Response: Information updated.
Comment: English should be edited by native speaker who are familiar with this field.
Response: The publication was edited by a native speaker.
For manuscript, please see the attachment.
Round 2
Reviewer 2 Report
Reviced well. Only the following point may be edited by editorial office.
Page 2. line 55; "for the very firats time the time"→"for the very first time".